# Function and Evolution of C1-2i Subclass of C2H2-Type Zinc Finger Transcription Factors in POPLAR

**DOI:** 10.3390/genes13101843

**Published:** 2022-10-12

**Authors:** Ping Li, Anmin Yu, Rui Sun, Aizhong Liu

**Affiliations:** Key Laboratory for Forest Resource Conservation and Utilization in the Southwest Mountains of China (Ministry of Education), Southwest Forestry University, Kunming 650224, China

**Keywords:** ZAT, *Populus*, ZnF-C2H2, stress response, evolution

## Abstract

C2H2 zinc finger (C2H2-ZF) transcription factors participate in various aspects of normal plant growth regulation and stress responses. C1-2i C2H2-ZFs are a special subclass of conserved proteins that contain two ZnF-C2H2 domains. Some C1-2i C2H2-ZFs in *Arabidopsis* (ZAT) are involved in stress resistance and other functions. However, there is limited information on C1-2i C2H2-ZFs in *Populus trichocarpa* (*PtriZATs*). To analyze the function and evolution of C1-2i C2H2-ZFs, eleven *PtriZATs* were identified in *P. trichocarpa*, which can be classified into two subgroups. The protein structure, conserved ZnF-C2H2 domains and QALGGH motifs, showed high conservation during the evolution of *PtriZATs* in *P. trichocarpa*. The spacing between two ZnF-C2H2 domains, chromosomal locations and cis-elements implied the original proteins and function of *PtriZATs*. Furthermore, the gene expression of different tissues and stress treatment showed the functional differentiation of *PtriZATs* subgroups and their stress response function. The analysis of C1-2i C2H2-ZFs in different *Populus* species and plants implied their evolution and differentiation, especially in terms of stress resistance. Cis-elements and expression pattern analysis of interaction proteins implied the function of *PtriZATs* through binding with stress-related genes, which are involved in gene regulation by via epigenetic modification through histone regulation, DNA methylation, ubiquitination, etc. Our results for the origin and evolution of *PtriZATs* will contribute to understanding the functional differentiation of C1-2i C2H2-ZFs in *P. trichocarpa.* The interaction and expression results will lay a foundation for the further functional investigation of their roles and biological processes in *Populus*.

## 1. Introduction

Zinc-binding domains were first identified in the protein TFIIIA of *Xenopus oocytes*, and function in the binding of IIA and DNA with cysteines and histidines in the enrichment folding domain centered on a zinc ion [1]. Zinc finger proteins (ZnFs) are the abundant proteins in eukaryotes, with diverse functions, including DNA recognition, RNA packaging, transcription regulation, protein folding, and assembly [2]. Structure studies of ZnFs revealed their diversity, which can be classified into classical, GATA, RanBP and A20 ZnFs according to their zinc ligation topology [3]. Classical ZnFs include a short β hairpin and α helix with a zinc atom coordinated with Cys-Cys-His-His or Cys-Cys-His-Cys subunits [3]. Most classical ZnF Cys2His2 zinc fingers are transcription factors that provide DNA binding and transcription regulation via their ββα framework of a Cys2His2 (C2H2) zinc finger module [2]. The stability of the C2H2 zinc finger ββα framework is maintained by the interlaced linkage provided by Zn^2+^, which binds with two pairs of histidines at the C-terminus of the α-helix and two cysteines at the end of the β-strand [4]. RS_CRZ1, a C2H2 ZnF of the tomato plant, is involved in the hostile environment encountered during *Rhizoctonia solani* host colonization [5].

The function of C2H2 ZnFs in growth regulation has been reported in plants. POPOVICH (POP), which encodes a C2H2 zinc-finger transcription factor (C2H2 ZTF), plays an important function during the evolution of *Aquilegia* nectar by regulating cell proliferation during the early phase of spur development [6]. VviZEPs may be a candidate regulator of pollen and grapevine reproductive development [7]. Hair, which encodes C2H2 ZnF, regulates trichome formation in Solanaceae species, where knockout and overexpression can directly repress or trigger trichome formation [8]. C2H2 ZTF encodes SE3.1, which is involved in the transition process from stigmas exsertion to insertion and affects the rate of self-fertilization in tomatoes [9]. Non Stop Glumes1 (NSG), encodes C2H2 ZTF, which can bind to Leafy Hull Sterile1 (LHS1) and recruits the corepressor TOPLESS-related protein to repress expression by downregulating histone acetylation levels of the chromatin, and then maintains an organ identified in the spikelet of *Oryza sativa* [10].

Stress resistance is another recognized function that involves C2H2. *TaZFP1B,* a C2H2-ZEP type transcription factor gene of wheat, can regulate P acquisition, ROS homeostasis and root system establishment to enhance plants’ Pi deprivation tolerance [11]. Overexpression of *TaZFP1B* can stimulate plant growth and oxidative resistance, which enhances wheat drought tolerance at critical periods [12]. *PeSTZ1*, a C2H2 ZTF coding gene of *Populus euphratica*, directly regulates PeAPX2 (ascorbate peroxidase of *P. euphratica*) to modulate plant ROS scavenging and enhances the freezing tolerance of poplar [13]. *GhSTOP1*, a C2H2 ZTF of *Gossypium hirsutum* L., can impact *GhALMT1* (Aluminum-activated Malate Transporter 1), *GhMATE* (Multidrug and Toxic Compound Extrusion) and *GhALS3* (Aluminum Sensitive 3) to regulate aluminum and proton stress tolerance of cotton [14]. MpZEF1, a C2H2 ZTF of *Millettia pinnata*, can increase the accumulation of stress-responsive genes and reduce reactive oxygen species (ROS) to enhance plant salt stress resistance [15]. Except for the influence on stress responses, C2H2 ZTFs may regulate the catechin accumulation of *Camellia sinensis* (L.) O. Kuntze [16]. 

Based on the protein pattern, all the C2H2 ZTFs of *Arabidopsis thaliana* were divided into three sets (A, B and C), and each set was further divided into different families as C1, C2 and C3 in set C [17,18]. Family C1 comprised conserved zinc finger helices with an invariant QALGGH motif, which can be subdivided into C1-1i, C1-2i, C1-3i, C1-4i, and C1-5i according to the number of isolated fingers [18]. The 20 C1-2i members of *A.thaliana* were assigned into five subgroups (C1-2iA, C1-2iB, C1-2iC, C1-2iD, and C1-2iX) according to sequence similarity [18]. ZATs (Zn transporter of *A. thaliana*) were a subtype of C1-2i Classical ZnFs Cys2His2 zinc fingers, whose function was unclear [18]. The four ZATs of the C1-2iD subgroup were involved in water stress response and salt resistance, which can be speculated from their expression change [19]. Van et al. isolated *ZAT* genes and found its 398 amino acid residues predicted six membrane-spanning domains in transgenic plants that can enhance Zn resistance and maintain Zn homeostasis [20]. GmZAT4, a typical C2H2 ZTF of soybean, can enhance plant osmotic and salt stress resistance and ABA responses through overexpression [21]. *MdZAT10*, a C2H2 ZTF gene of the‘Royal Gala’ apple and a homolog of *Arabidopsis* ZAT10, was negatively regulated for the drought tolerance of the apple by inducing a higher malondialdehyde (MDA) content and higher reactive oxygen species (ROS) [22]. ZATs can be affected by DNA methylation, whose methylation level was significantly greater in adult leaves than juvenile leaves, which limited their function in the mature period in *Malus hupehensis* [23]. Seven ZATs were discovered in *M. hupehensis* [23]. 

Although many studies have found important functions of C2H2 ZTF, a comprehensive investigation of ZATs (C1-2i C2H2-type zinc finger transcription factors), which are an important subtype of C2H2 ZTF, has not yet been completed. In the present study, we will investigate the structure and predict the function of ZATs in *P. trichocarpa* and different *P.* species. We first identified eleven PtriZATs in *P. trichocarpa*. The phylogenetic relationship, two conserved ZnF-C2H2 domains, motifs, gene structure, and cis-elements were characterized. The expression pattern of *PtriZAT*s in different growth stages and stress resistance were analyzed. The distribution and structure of ZATs in different *Populus* species were also analyzed. Accordingly, our results provide information for ZATs in *Pupulus* and suggestions for further function investigation. 

## 2. Materials and Methods 

### 2.1. Identification of PtriZAT Members in P. trichocarpa and Populus Speciecs

To identify members of ZAT in *P. trichocarpa*, a sequence similarity search was performed using the *Arabidopsis* ZAT proteins with BLASTP of the standalone BLAST+ tool (NCBI-blast-2.7.1+) [18]. Seventeen proteins were obtained with similarity to *A. thaliana* ZATs (E-value ≤ 0.05). Protein sequences of identified ZAT homologs in *P. trichocarpa* were subjected to domain analysis using PFAM (https://pfam.xfam.org/ (accessed on 19 January 2022)), SMART (http://smart.embl-heidelberg.de/ (accessed on 27 January 2022)) and Web CD-Search Tool (https://www.ncbi.nlm.nih.gov/Structure/bwrpsb/bwrpsb.cgi (accessed on 17 January 2022)). According to the structure of ZAT domains, eleven proteins were analyzed as ZATs in *P. trichocarpa.* ZATs in different *Populus* were obtained by BLASTP with protein sequences of PtriZATs. The genome data of *Populus deltoids*, *P. euphratica*, *Populus alba* and *Ptomentosa tomentosa* were acquired from NCBI (https://www.ncbi.nlm.nih.gov/ (accessed on 24 January 2022)). The subcellular localization of PtriZATs was predicted by WOLF PSORT (https://wolfpsort.hgc.jp/ (accessed on 24 January 2022)). The features of proteins were analyzed using Expasy-ProtParam (https://web.expasy.org/protparam/ (accessed on 26 January 2022)).

### 2.2. Phylogenetic Relationship of ZAT Proteins in P. trichocarpa

The sequences of 11 ZAT proteins in *P. trichocarpa* were obtained from Phytozome13 (https://phytozome-next.jgi.doe.gov/ (accessed on 20 January 2022)). To analyze the phylogenetic relationship between different ZATs in *P. trichocarpa*, their full-length protein sequences were aligned using ClustalW [24]. The phylogenetic tree of ZAT proteins was constructed using MEGA11 with the Maximum Likelihood (ML) method [25]. The bootstrap values reported for each branch reflected the percentage of 1000 replicate trees containing that branch. 

### 2.3. Exon Intron Structure, Location, Conserved Motifs and Promoter in PtriZAT Genes

The exon intron structures of eleven *PtriZAT* genes were analyzed using the Gene Structure View package of TBtools (v1.098691) with geneID and gff3 data of *P. trichocarpa* [26]. The chromosomal location of all identified *PtriZAT* genes on the chromosome was mapped using Chromosome Map Tool Blast, Text merge for MCScanx, Quick run MCScanX Wrapper and circle gene view packages of TBtools (v1.098691). The conserved motifs in PtriZAT proteins were searched using Multiple Em for Motif Elicitation (MEME) (https://meme-suite.org/meme/tools/meme (accessed on 19 January 2022)). The conserved cis-elements in the promoters of *PtriZAT* genes were analyzed using the PLACE database (https://www.dna.arc.go.jp/PLACE/ (accessed on 19 January 2022)) with the 2 kb sequences upstream of the translational start codon. 

### 2.4. Plant Materials and qRT-PCR Assays

The poplars were gathered from Kunming (E102.74°, N25.17°), and well grown in the culture room of Southwest Forestry University, Kunming, with the cutting method and natural culture conditions. The fresh and healthy leaves, stems, and roots were collected. For drought, ABA, and salt treatment, there was a no-watering protocol (seven days), and 10 μM ABA and 250 Mm NaCl were added into the normal condition. Total RNAs were extracted using the RNAprep Pure Plant Plus Kit (Cat. DP441, Tiangen, Beijing, China) following the manufacturer’s instructions for different tissues and stress treatment plant materials. RNA (1 μg) was used for reverse transcription with an EasyScript^®^ All-in-One First-Strand cDNA Synthesis SuperMix for qPCR reagent Kit (Transgene, Beijing, China). The relative expression levels of individual genes were measured with gene-specific primers (Appendix A) by real-time quantitative PCR (qRT-PCR) analysis, which was carried out in a 20 μL reaction mix with 1 μL of diluted cDNA template and TransStart^®^ Green qPCR SuperMix (Transgene, Beijing, China) with a Bio-Rad CFX96. The elongation factor 1 (EF1) served as the internal control [27].

### 2.5. Expression Profiles of PtriZAT Genes in Different Plants

*P. trichocarpa* and *Physcomitrium patens* transcriptome data were obtained from Phytozome 13 (https://phytozome-next.jgi.doe.gov/ (accessed on 20 January 2022)). The expression data of *PtriZAT* genes were used to analyze the expression profile of *P. trichocarpa* H1 genes at different development stages (female, male, bud, leaf, root, and stem) and treatments (ABA, ACC, BAP, BL, GA, NAA, SA, SL, and meJA). All the expression values demonstrated in the heatmap were calculated with log2 of the FPKM values of Phytozome 13. The heatmap of *PtriZAT* genes was obtained by Heml 1.0.3 [28].

### 2.6. Prediction of the Interaction Proteins of PtriZATs

The potential interaction proteins of PtriZATs were predicted using an online STRING server (https://string-db.org (accessed on 17 March 2022)) with the protein sequences of PtriZATs. All the interaction proteins of different PtriZATs were examined using contrastive analysis; the number and messages of corresponding proteins are listed in Appendix A. The function annotation of interaction proteins was obtained in Phytozome 13.

## 3. Results

### 3.1. Genome-Wide Identification of PtriZAT Members in P. trichocarpa

After a sequence similarity search in *P. trichocarpa* with the 20 C1-2i C2H2 ZTF members of *A. thaliana*, 17 proteins of *P. trichocarpa* were obtained (Appendix A). Of the 17 candidate proteins, only 11 members that were conserved contained two ZnF_C2H2 domains, which was the specific feature of C1-2i subclass ZFs (Table 1, Figure 1A [18]). The subcellular localization of 11 *PtriZATs* proteins predicted by WOLF PSORT showed their nucleus location (https://wolfpsort.hgc.jp/ (accessed on 24 January 2022)). The length of *PtriZATs* proteins varied from 179 to 310 amino acids. The length of Zn-C2H2 domains also varied from 24 to 26 amino acids. All the *PtriZATs* were neutral or basic proteins, which were aliphatic with low hydropathicity (Table 1). 

To compare with other plants, we performed a homology search for typical plants using *Arabidopsis* ZATs through forward- and reverse-comparison, and confirmed the ZAT proteins with two Zn-C2H2 domains using the Web CD-Search Tool. *Carya illinoinensis*, *Oryza sativa*, *Lactuca sativa*, *Manihot esculenta*, *Medicago truncatula*, *Solanum Tuberosum*, and *Zea mays* contained more ZATs than *P. trichocarpa*. *Ceratodon purpureus*, *β vulgaris* L., *Theobroma cacao*, and *Vitis vinifera* had a small number of ZATs (Appendix A).

### 3.2. Sequence Characteristics and Phylogenetic Relationships of PtriZATs

To obtain the sequence characteristic of *PtriZATs*, all 11 proteins were aligned and analyzed using BioEdit version 7.0.9.0 [29]. All the *PtriZATs* had two conserved ZnF-C2H2 domains with a specific plant QALGGH motif [18]. The 11 *PtriZATs* can be classified into 2 subgroups according to their sequence characteristics of the ZnF-C2H2 domain (Figure 1A). Seven *PtriZATs* (Potri.004G216900, Potri.006G121600, Potri.008G032300, Potri.010G229400, Potri.010G209400, Potri.001G235800, and Potri.009G027700) were conserved and had Lys134, Thr135, Arg150, and Lys154 during the first ZnF_C2H2 domain and Gly218, Glu220 (Val220 in Potri.008G032300), Met231, Arg233, and Arg235 during the second ZnF_C2H2 domain, which belonged to subgroup I. The other four PtriZATs (Potri.001G295500, Potri.009G089400, Potri.002G119300, and Potri.014G017300) replaced the conserved subunit with Thr/Ser134, Val135, Lys150, Arg154, His218, Thr220, Lys231, Cys233, and Tyr235, which belonged to subgroup II. The different subunits of *PtriZATs* did not influence the protein structure. All the *PtriZATs* contained two ZnF-C2H2 domains, which were ββα frameworks with a linkage interlaced by Zn^2+^ (Figure 1B,C).

According to the characteristic bases of C1-2i ZTFs of *A. thaliana*, *PtriZATs* belonged to the C1-2iC and C1-2iD subgroups [18]. The subgroup I *PtriZATs* were conserved and contained Arg138, Ser142, Phe143, Arg50, Lys54, and Arg/Lys55 in the helix of the first ZnF-C2H2 domain, and Gly224, Met231, Arg232, Arg233, and Arg235 in the helix of the second ZnF-C2H2 domain, which was characteristic of C1-2iC ZEPs of *A. thaliana*. The subgroup II *PtriZATs* substitute contained Lys138, Ser142, Tyr143, Lys50, Arg54, and Lys55 in the helix of the first ZnF-C2H2 domain, and Lys219, Thr223, Gly224, Lys231, Arg232, Cys233, and Tyr235 in the helix of the second ZnF-C2H2 domain, which was characteristic of C1-2iD ZEPs of *A. thaliana* (Figure 1A).

To understand the phylogenetic relationship of *PtriZATs*, a phylogenetic tree was built with sequences of PtriZAT proteins. The phylogenetic tree of *PtriZATs* revealed seven proteins that belonged to the same branches, consistent with the analysis results for protein structure (Figure 1 and Figure 2A). The other four *PtriZATs* belonged to the other branch with longer footsteps. To analyze the phylogenetic relationship of *PtriZAT*s with subclass C1-2i ZEPs of *A. thaliana*, an unrooted tree was constructed with their protein sequences. Except for the C1-2iD subgroup ZEPs, the other *A. thaliana* ZEPs were separated into an independent branch (Appendix A). The phylogenetic tree of ZATs of a different plant also showed the two conserved subtypes of *PtriZATs*, which were sorted into two different branches with most ZATs of different plants, which may be according to different evolutionary relationships and sequence variation (Appendix A).

### 3.3. Gene, Protein Structure, and Conserved Motifs of PtriZAT Genes

To further understand the structure of *PtriZATs*, we searched 15 conserved motifs in *PtriZATs* with the MEME software. As shown in Figure 2B, motif1, motif2, and motif3 were the most conserved motifs, which were identified in all the *PtriZATs*. Subgroup II *PtriZATs* had seven conserved motifs (motif 1, motif 2, motif 3, motif 4, motif 5, motif 6, and motif 8). *PtriZATs* of subgroup I can be classified into two subtypes, Potri.004G216900, Potri.006G121600, Potri.008G032300, and Potri.010G229400 were the first subtype, which conservedhad motif1, motif2, motif3 and motif5. Potri.010G209400, Potri.001G235800 and Potri.009G027700 were the second subtype, which when conserved had motif 1, motif 2, motif 3, motif 4 and motif 7. The *PtriZAT*s of subgroup II were conserved and had motif 1, motif 2, motif 3, motif 4, motif 5, motif 6 and motif 8 (Appendix A). Protein structure analysis of *PtriZATs* found two conserved ZnF-C2H2 domains, the locations of which differed according to the subgroups (Figure 2C). The first subtype, *PtriZATs* of subgroup I, had longer sequences, with ZnF-C2H2 domains near the C-terminal. Meanwhile, the second subtype, *PtriZATs* of subgroup I, had the shortest sequences. To reveal the coding characteristic of *PtriZATs*, the gene structure was investigated by comparing the CDS sequences. The results showed that all *PtriZAT* genes were conserved and had a single exon, except for Potri.006G121600, which contained two exons linked with a short intron and without UTR sequences (Figure 2E). 

### 3.4. Chromosomal Location and Duplication Events of PtriZAT Genes

To investigate the possible relationship between *PtriZATs*, we analyzed their chromosomal location using TBtools (Figure 3). We found eleven *PtriZATs* genes distributed on eight chromosomes randomly. No *PtriZAT*s were located on chromosome 3, 5, 7, 11, 12, and 13. Except for Potri.001G235800, Potri.001G295500, Potri.009G027700, Potri.009G089400, Potri.010G229400, and Potri.010G209400, other *PtriZAT*s were located on chromosome 2, 4, 6, 8, and 14 separately. Potri.008G032300, Potri.014G017300, Potri.004G216900, Potri.010G229400 and Potri.010G209400 were located on the distal ends of chromosomes 8, 14, and 10. The distribution of *PtriZAT* genes relative to the corresponding duplication is shown in Figure 3B. Potri.008G032300 was probably the primordial *PtriZAT*, which replicated to Potri.004G216900, Potri.006G121600, and Potri.010G229400 with genome duplication [30]. The other *PtriZAT*s with synteny contained conserved ZnF-C2H2 domains and belonged to the same subgroup, which probably developed because of genome duplication [30].

### 3.5. Cis-Elements in the Promoter Regions of PtriZAT Genes

To investigate the regulation pattern, we detected cis-elements in the promoter regions of *PtriZAT* genes (Figure 2D, Appendix A). Many cis-elements related to stress response and growth regulation were predicted by PlantCARE. The most frequent cis-elements were G-Box, which were detected in ten *PtriZATs,* except Potri.001G235800, which suggested the important function of light signals during transcription regulation. ABRE was the other widespread cis-elements in *PtriZATs*, which participated in the response of abscisic acid (ABA), which indicates the function of *PtriZAT*s during stress response and growth regulation. The TGACG-motif, TC-rich repeats, LTR, AACA_motif, P-box, MBS, TCA-element, AuxRR-core, and WUN-motif, predicted in *PtriZATs*, function in response to MeJA, defense, low-temperatures, gibberellin, drought-inducibility, salicylic acid and auxin, which were important for plant growth under stress. Other growth regulation cis-elements, such as circadian, CAT-box, RY-element, GC-motif, AT-rich sequence, GCN4_motif, and MSA-like, were also detected, and involved in functions for circadian control, meristem expression, seed-specific regulation, anoxic specific inducibility, maximal elicitor-mediated activation, endosperm expression, and cell cycle regulation. In addition, a zinc metabolism regulation cis-element, O_2_-site, was detected in subgroup I *PtriZATs*, which implied the function of Zn^2+^ in *PtriZATs*.

### 3.6. Expression Patterns of PtriZAT Genes in Different Tissues and Stress Treatment

To investigate the function of *PtriZAT* genes the transcript abundance of *PtriZAT*s were analyzed based on the transcription data of four growth stages, nine stress treatments, and plants of two sexes from Phytozome 13 (Appendix A). All *PtriZAT*s were lowly expressed in all the growth stages and induced by stress treatment. The expression pattern of ten expressed *PtriZAT*s was validated with qRT-PCR. The primer sequences for qRT-PCR are listed in Appendix A. Based on the qRT-PCR analysis (Figure 4), the selected *PtriZAT*s were highly induced by stress, especially ABA and salt. Notably, the significantly upregulated *PtriZAT*s under abiotic stress, suggested their positive regulation, which may relate to the stress- response cis-elements (Figure 2 and Figure 4). Except for stress treatment, the expression of most *PtriZAT*s maintained a low level during different tissues. In addition, the expression of some *PtriZAT*s varied in different tissues, which may be because of the growth-regulation cis-elements (Figure 2 and Figure 4). At the same time, the expression pattern of some subgroup I *PtriZAT*s (Potri.010G209400 and Potri.002G119300) were significantly induced by growth tissues such as root and stem compared to stress, which may relate to the functional complementarity caused by gene duplication events (Figure 3 and Figure 4). Interestingly, a subgroup II *PtriZAT* (Potri.002G119300) highly induced both growth tissues and stress, which was different from the variation of the expression of other *PtriZAT*s. The signal and process of *PtriZAT*s involved need more research.

### 3.7. Protein–Protein Interaction Networks

C2H2 ZFs may form homodimers with other transcription factors and proteins in Drosophila with long-distance interaction between the activator and reporter gene promoter, which influences protein structure and functions [31]. Protein–protein interaction network analysis was completed to investigate the regulation mechanisms of *PtriZATs*. The predicted protein–protein interaction of *PtriZATs* based on *P. trichocarpa* is shown in Figure 5. Some PtriZATs interacted with other *PtriZATs* (Appendix A). Except for that case, all *PtriZATs* interacted with node 1 (Potri.008G195900), node 2 (Potri.005G246700), node 3 (Potri.011G168800), node 5 (Potri.001G315200), node 7 (Potri.010G019150), node 8(Potri.013G006000), node 10 (Potri.017G055200), and node 11 (Potri.018G084900). CHR910 (Potri.008G195900) was a nucleoplasmin ATPase, which is functional on the DNA helicase and histone interaction in plants [32]. HMG 20 (Potri.005G246700, Potri.011G168800) was reported to have a potential role for β-dystrobrevin in neuronal differentiation, and histone deacetylase activity regulation [33,34]. PKGII (Potri.018G084900) inhibited the proliferation of various cancer cell lines in humans [35,36,37]. MORPHEUS (Potri.013G006000) MORPHEUS (Potri.013G006000) was a plant-specific epigenetic regulator of transcriptional gene silencing and affected cytosine methylation status [38]. SWI/SNF-Related proteins (Potri.005G246700, Potri.011G168800, Potri.010G019150) regulated the progression of meiosis in male reproductive cells [39]. UFD1 (ubiquitin fusion degradation protein 1, Potri.004G163200) involved in the ubiquitin-proteasome system may interact with *PtriZAT*s [40]. Based on the functional annotation of homologous proteins, it is possible to predict the chromatin regulation, transcriptional gene silencing, and ubiquitination- regulation function of *PtriZAT*s.

To investigate the function of the interaction proteins of PtriZATs, we analyzed their cis-elements and expression patterns under different growth stages and stress. The specific interaction protein of Potri.010G209400, Potri.009G124900, was functional for growth regulation and all stress responses of *P. trichocarpa.* Potri.010G209400, a PtriZAT of subgroup I, functioned in a significant response of ABA, ACC, BAP, GA, NAA, SA, and meJA and may be related to Potri.009G124900 (Figure 6, Appendix A). Most interaction proteins had no explicit comments. The cis-elements with the 2000 bp upstream of these cis-elements included many stress-response binding motifs, such as MBS (MYB binding site involved in drought-inducibility), TATC-box (gibberellin responsiveness), LTR (low-temperature responsiveness), TCA-element (salicylic acid responsiveness), ABRE (abscisic acid responsiveness), ARE (anaerobic induction), AuxRR-core (auxin responsiveness), TGACG-motif (MeJA responsiveness), P-box (gibberellin responsiveness) and WUN-motif (wound responsiveness) (Figure 7, Appendix A). 

### 3.8. The Distribution of PtriZATs in Populus Species

To investigate the distribution of ZATs in different *Populus* species, ZATs with two conserved ZnF-C2H2 domains in different *Populus* species were obtained by BLASTP using *PtriZAT*s. The results show that different ZATs were distributed in different *Populus* species, and 16 ZATs were in *P. deltoids*, 19 ZATs were in *P. euphratica*, 29 ZATs were in *P. alba*, and 38 ZATs were in *P. tomentosa* (Appendix A). The sequence alignment of ZATs in different *Populus* species had variable subunits in conserved QAGGH ZnF-C2H2 domains (Appendix A, [18]). Total sequence variation of QAGGH was obtained in Podel.01G169100.1.p, Podel.10G160600.1.p, and Podel.17G095700.1.p of *P. deltoids*; XP_011001488.1, XP_011012172.1, and XP_011031881.1 in *P. euphratica*; XP_034907133.1, XP_034922750.1, and XP_034932650.1 in *P. alba*, KAG6741799.1, KAG6742946.1, KAG6757992.1, KAG6759880.1, KAG6780647.1, KAG6782363.1, KAG6788764.1, KAG6789197.1, KAG6792151.1 and KAG6792606.1 in *P. tomentosa.* Some ZATs substituted Ala for Gly of QAGGH, such as KAG6755684.1 and KAG6756859.1 in *P. tomentosa.* All the variable ZATs in *Populus* had development branches with subgroup I and subgroup II *PtriZAT*s (Appendix A). The other *Populus* ZATs were divided into two subgroups with *PtriZAT*s (Appendix A).

## 4. Discussion

### 4.1. Characteristics of PtriZATs in P. trichocarpa

We identified eleven *PtriZATs* with two conserved ZnF_C2H2 domains in *P. trichocarpa* (Figure 1, Appendix A). Based on the phylogenetic tree of eleven *PtriZATs*, we classified them into two subgroups (Figure 2). A comparison of the protein sequences and structures revealed the variation of *PtriZATs*. The length of the *PtriZATs* varied according to subgroups. The length of ZnF-C2H2 domains of subgroup I *PtriZATs* (25–27 amino acids) was longer than subgroup II (24–25 amino acids) (Table 1). The location of ZnF-C2H2 domains in *PtriZATs* also varied according to the subgroups. The two ZnF-C2H2 domains were closer to the C-terminal in the first subtypes *PtriZATs* of subgroup I than subgroup II (Table 1, Figure 2). The length of spacing between the two ZnF-C2H2 domains varied from 19 to 54 amino acids, and the amino acid subunits also varied in different *PtriZATs*. The spacing between the core motifs of ZnF-C2H2 might be important for target sequence recognition [41], and there may be some difference in DNA-binding between different *PtriZATs*. The length of PtriZATs of subgroup I was also longer than subgroup II. Individual fingers of C2H2-ZF domains are specifically bound over a wide range of three base pair targets [42]. The higher quantity of ZnF-C2H2 domains and the ZnF-C2H2 DNA-binding landscapes confirmed the important role of C2H2 in the DNA binding of longer DNA sequences [43]. The length of ZnF-C2H2 domains of *PtriZATs* ensured the stability of a Zn finger and DNA binding ability, as reported previously [44]. The variety of spacers between the two fingers provided the necessary gaps between the core sites of the target DNA [18]. The highly conserved QALGGH motifs of *PtriZATs* were the same as Family C1 C2H2 of *Arabidopsis* and other plants, which were conserved zinc finger helices for DNA binding [18,41]. 

### 4.2. Different Distributions of ZATs in Populus Species

Different numbers of ZATs were obtained in different *Populus* (Appendix A). The different distributions of ZATs in different *Populus* and plants might reflect their variation in function and evolution in *Populus* and plants (Appendix A). To investigate the evolutionary history of ZATs in different *Populus* and plants, a phylogenetic tree was built using protein sequences (Appendix A). Seven *P. deltoids* ZATs, seven *P. euphratica* ZATs, five *P. alba* ZATs and fourteen *P. tomentosa* ZATs were separated into subgroup I with subgroup I PtriZATs. Nine *P. deltoids* ZATs, seven *P. euphratica* ZATs, six *P. alba* ZATs, seven *P. tomentosa* ZATs were separated into subgroup II with subgroup II PtriZATs. The separation of most *Populus* ZATs showed the conservation of ZATs in *Populus*. To investigate the difference in independent ZATs in *Populus*, sequence analysis was conducted using *Populus* ZATs. All the independent ZATs contained variation motifs in the Zn-C2H2 domain (Appendix A). C2H2 ZTFs, one of the largest gene families, were conserved in flora and fauna, the evolution of base-contacting residues for DNA binding, and the interaction of non-base-contacting residues with a DNA backbone, which plays a key role during their expansion in metazoans [45]. Sequence variation of QAGGH motifs obtained for different *Populus* species reflected the evolutionary events of *Populus*, especially the residue substitute of Gly (Appendix A). The hydrophobic amino acids surrounding the motifs of ZnF-C2H2 were involved in protein–protein interaction and might interact with other DNA-binding proteins [41,46]. The total sequence variation of QAGGH in *Populus* species might imply larger evolutionary events and different DNA-binding functions in *Populus* C2H2-ZnFs (Appendix A).

### 4.3. Chromosomal Distribution and Gene Duplication of PtriZATs

According to the chromosome location results, Potri.004G216900, Potri.006G121600, and Potri.010G229400 were possibly derived from Potri.008G032300 (Figure 3). All four *PtriZATs* belonged to the first subtype of Subgroup I with conserved ZnF-C2H2 domains and amino acid subunits (Figure 1 and Figure 2). The detected higher-expressed Potri.004G216900 under ABA treatment in the first subtype of subgroup *PtriZATs* may relate to the evolution and origin of these *PtriZATs* (Figure 4, Appendix A). Previous studies have identified that the *Populus* genome underwent at least three rounds of genome-wide duplication, followed by multiple segmental duplications, tandem duplications, and transposition events, such as retrotransposition and replicative transposition [30,47]. The origin of the four *PtriZATs* in the first subtype of subgroup ZATs may coincide with the genome-wide duplication. The location relationship of four Subgroup II *PtriZATs* was also a reflection of the genome-wide duplication, which was equalization pipeline-connected. However, the origins of other *PtriZAT*s were unclear.

### 4.4. Functional Differentiation of PtriZATs

Most ZFPs take part in plant growth regulation and stress response. The ZFPs in *G. hirsutum* are involved in fiber cell growth and hormone response [48]. The faster and earlier response of *PtaZFP2* monitored during stem bending and stress response implied its function in the growth and environmental adaption of *Populus* [49]. PeSTZ1, a C2H2-type zinc finger of *P. euphratica*, can regulate PeAPX2 to modulate ROS scavenging and enhance the freezing tolerance of *Populus* [13]. The higher expression of *PtriZAT*s of subgroup II and the second subtype of subgroup I in all the stress treatments reflected the importance and wide-spectrum function of *PtriZATs* under stress (Figure 4 and Appendix A). The motifs and cis-elements related to phytohormone and abiotic stress resistance identified in *PtriZATs* indicated their stress resistance and growth regulation functions (Appendix A; [47]). Combining the expression level and cis-element analysis showed that *PtriZATs* take part in the ABA, meJA, and light responses of plants. The higher expression level of *PtriZATs* in male plants may imply a different function of *PtriZATs* during sex differentiation. In *P. yunnanensis* Dode, there were many sex-specific responses in growth regulation and physiological metabolism [50]. Under a stress condition, *Populus* also shows a sex-specific response in physiological, morphological, proteome, and gene expression levels to heavy metals, salinity, drought, and nutrient deficiency [51]. During growth and stress resistance, males of *Populus* were more adaptive [52]. The higher expression of *PtriZATs* in male *Populus* was consistent with the strength of male plants. The interaction proteins of *PtriZATs* support growth regulation and the stress response function, which may relate to epigenetic mechanisms, such as histone modification [33,34], cytosine methylation [38], and ubiquitination [40]. The highly induced Potri.006G192700 and Potri.016G045500 in male plants provided the specific function of *PtriZATs* during sex differentiation (Figure 6, Appendix A), which may be because of its function in the progression of meiosis in the male reproductive cell of SWI/SNF-Related proteins [39]. Cis-element interaction proteins contained and had a higher induced expression of *PtriZATs* genes, implying a function in the stress response of *PtriZATs*, which may occur through interaction with stress-related genes (Figure 4 and Figure 7, Appendix A). Except for the functional interaction proteins of PtriZATs reported in plants, the function of other interaction proteins with a role in animals and diseases needs more research in plants [53,54] (Appendix A).

Overall, the study results indicate that the *PtriZATs* functions were the same as the functions for DNA recognition, transcription regulation, protein folding and assembly of ZnFs in eukaryotes [2], but the mechanisms in plants need further research.

## 5. Conclusions

This study provided a thorough overview of the C1-2i subclass of C2H2-type Zinc Finger of *P. trichocarpa* and *Populus*. It presented a new perspective on the evolution and function of *PtriZATs*. The phylogenetic analysis found that eleven PtriZATs were classified into two subgroups according to their sequence characteristics of the ZnF-C2H2 domain. The phylogenetic relationship, exon-intron structure, chromosomal location, conserved motifs, and cis-regulatory elements of *PtriZAT*s were explored in this research. The phylogenetic relationship and exon intron structure analysis supported the two subgroups classification. Chromosomal mapping and collinearity analysis revealed the duplication events of *PtriZAT*s. Furthermore, the cis-elements analysis and transcription data of *PtriZAT*s revealed their special expression profiles focused on stress response and sex differentiation. The validation by qRT-PCR further illustrated the function of most *PtriZAT*s under stress, especially ABA and salt. The variation in the expression pattern of different PtriZATs may relate to functional complementation and gene duplication. The interaction proteins of PtriZATs predicted using STRING were functional as regulators of epigenetic mechanisms, which may provide clues for the functional study of *PtriZAT*s. Our study contributes to the understanding of the structure and role of growth tissues, *PtriZAT*s under stress, and provides resources for further functional analysis in *Populus*. It may contribute to improving the stress resistance of *Populus* using molecular biology techniques such as overexpression/mutation and regulation of epigenetic modification in future research.

## Figures and Tables

**Figure 1 genes-13-01843-f001:**
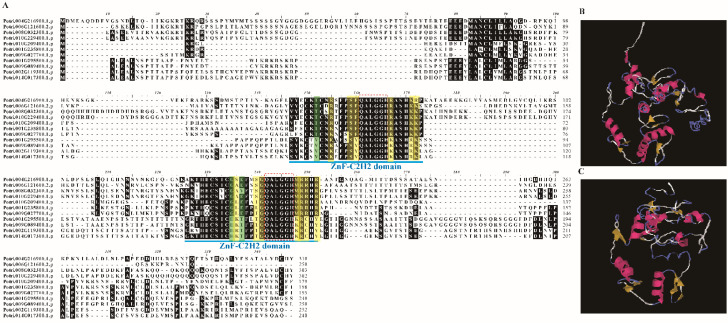
Sequence alignment and structure analysis of PtriZATs. (**A**). Multiple sequence alignment of PtriZAT proteins with BioEdit. Blue label and letters denote domains of PtriZAT proteins. The codes for shades are: Black, >50% conservation position with BioEdit; Green, same amino acid subunits; Yellow, characteristic bases of C1-2i ZEPs of *A. thalina* [18]. (**B**,**C**). Protein structure of ZnF-C2H2 domains in Potri.004G216900 and Potri.001G295500 using Expasy-ProtParam. Potri.004G216900 model was built using 6ml2.1.A template with 2.7Å X-ray method (0.30 seq similarity, 0.37 coverage, 120–233 range); Potri.001G295500 model was built using 6ml2.1.A template with 1.87Å X-ray method (0.30 seq similarity, 0.36 coverage, 63–148 range).

**Figure 2 genes-13-01843-f002:**
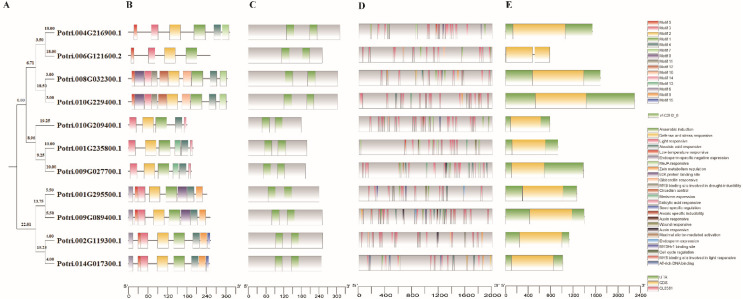
Phylogenetic tree, motifs and structure of *PtriZAT*s. (**A**). The phylogenetic tree of PtriZATs constructed on the basis of the maximum-likelihood method. (**B**). The conserved motifs of PtriZATs predicted using various Multiple Em for Motif Elicitation (MEME). The colored boxes represented 15 conserved motifs, the grey lines are non-conserved sequences. (**C**). A schematic diagram showing the ZnF-C2H2 domains of PtriZATs. The green boxes exhibited the ZnF-C2H2 domains. (**D**). Cis-elements of PtriZATs during the 2000bp upstream sequences. (**E**). Gene structures of PtriZAT genes. The yellow box showed the CDS, green ones were no-coding sequences.

**Figure 3 genes-13-01843-f003:**
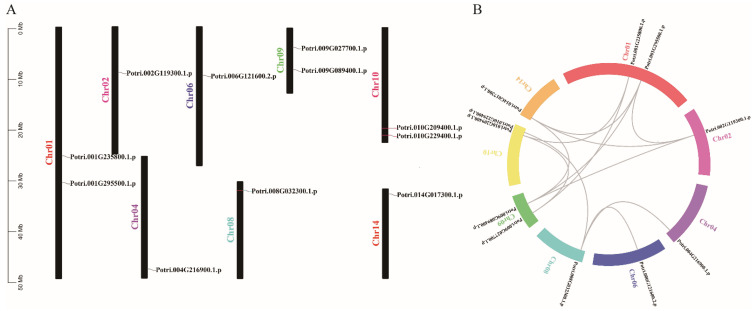
The chromosomal location and gene duplication events of *PtriZAT* genes. (**A**)**.** Chromosomal location of *PtriZAT* genes in *P. trichocarpa. PtriZAT* genes were widely mapped to 8 of the 19 *Populus* chromosomes. (**B**). The gene duplication events of *PtriZATs*. The paralogous pairs of *PtriZAT* gene in the segmental duplicated blocks were connected with gray lines. Chromosome numbers were indicated at the side of each line.

**Figure 4 genes-13-01843-f004:**
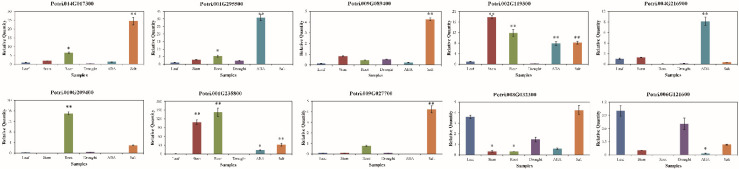
Expression of *PtriZAT* genes under stress treatment and different tissues. EF1 was used as an internal control. Data are means of three biological replicates and error bars are ± SE from three independent experiments, each performed with 2–3 leaves from three separate plants, asterisks indicate significant differences by Tukey LSD test (* *p* < 0.05, ** *p* < 0.01).

**Figure 5 genes-13-01843-f005:**
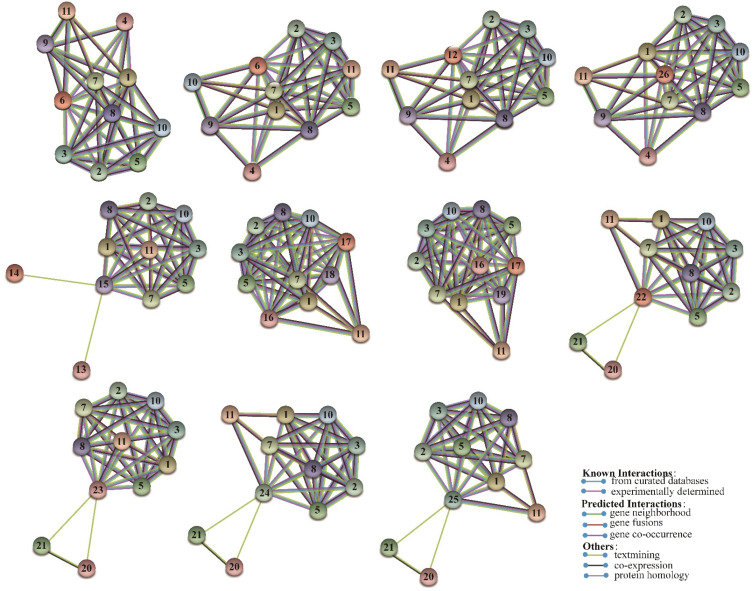
The predicted network of protein–protein interactions of PtriZATs by STRING. The different colors represent different types of interactions. The interaction proteins were as follows: 1, Potri.008G195900; 2, Potri.005G246700; 3, Potri.011G168800; 4, Potri.001G417300; 5, Potri.001G315200; 6, Potri.004G216900; 7, Potri.010G019150; 8, Potri.013G006000; 9, Potri.013G007800; 10, Potri.017G055200; 11, Potri.018G084900; 12, Potri.008G032300; 13, Potri.004G163200; 14, Potri.009G124900; 15, Potri.010G209400; 16, Potri.006G192700; 17, Potri.016G045500; 18, Potri.001G235800;19, Potri.009G027700; 20, Potri.001G143700; 21, Potri.001G173200; 22, Potri.001G295500; 23, Potri.009G089400; 24, Potri.002G119300; 25, Potri.014G017300; 26, Potri.010G229400.

**Figure 6 genes-13-01843-f006:**
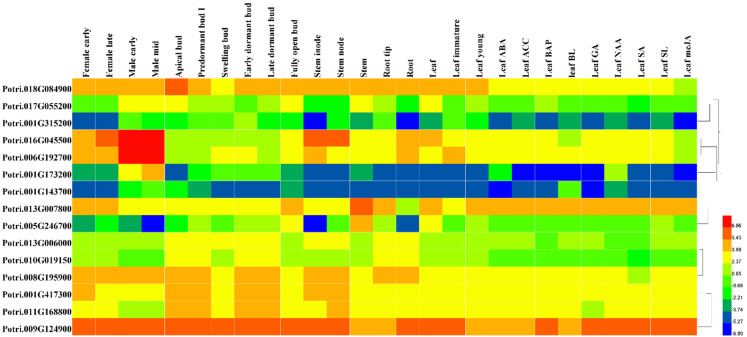
Expression of interaction proteins of PtriZATs encoded genes during different growth stages and stress response. The expression data of 15 PtriZAT genes were from Phytozome13. The sex differentiation samples including early and late growth of female and male plants. The samples of different tissues including bud (apical, predormant, leaf dormant, swelling, early dormant, fully open), leaf (young, immature), root (tip), steam (inode, node). The nine stress treatments include ABA, ACC, BAP, BL, GA, NAA, SA, SL and meJA. The color scale represents the value of Log2(FPKM).

**Figure 7 genes-13-01843-f007:**
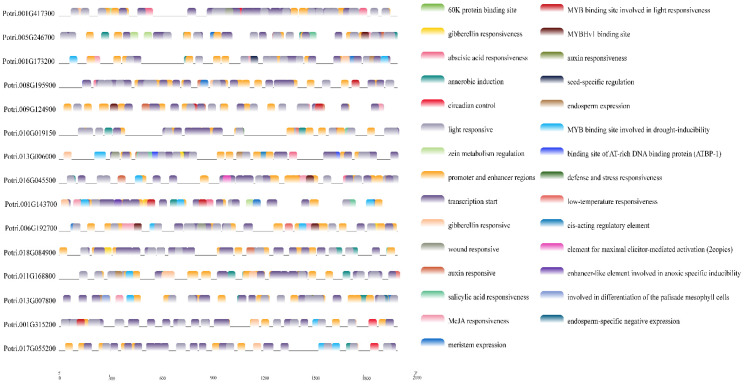
Cis-elements of *PtriZAT*s interaction proteins during the 2000 bp upstream sequences. The color labels were on the right, the cis-elements message seen on Appendix A.

**Table 1 genes-13-01843-t001:** Characteristics of PtriZAT proteins.

ID.	Number of Amino Acids	Zn-C2H2 Domain1	Zn-C2H2 Domain2	Molecular Weight	Theoretical pI	Aliphatic Index	Grand Average of Hydropathicity (GRAVY)	Location
Potri.001G235800.1.p	198	55–80	101–127	21,440.73	9.62	77.02	−0.399	nucle-
Potri.002G119300.1.p	252	93–117	146–171	27,162.32	7.7	59.68	−0.57	nucle-
Potri.004G216900.1.p	310	129–155	209–234	33,781.64	7.23	62.03	−0.695	nucle-
Potri.006G121600.2.p	250	184–209	113–139	27,202.53	9.01	63.24	−0.63	nucle-
Potri.008G032300.1.p	303	125–151	202–228	33,762.19	7.74	55.38	−0.938	nucle-
Potri.009G027700.1.p	193	51–76	99–123	21,035.39	9.57	76.89	−0.41	nucle-
Potri.009G089400.1.p	250	80–105	132–157	26,305.14	8.83	55.12	−0.657	nucle-
Potri.010G209400.1.p	179	46–72	91–116	20,059.96	9.06	61.62	−0.523	nucle-
Potri.010G229400.1.p	302	121–147	199–223	33,286.63	7.21	55.93	−0.901	nucle-
Potri.014G017300.1.p	248	91–115	146–171	26,605.77	8.4	58.27	−0.56	nucle-
Potri.001G295500.1.p	240	67–92	122–147	25,363.26	8.45	56.62	−0.577	nucle-

## Data Availability

Not applicable.

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
