# Peer review of "Function and Evolution of C1-2i Subclass of C2H2-Type Zinc Finger Transcription Factors in POPLAR"

_genes, 2022, doi:10.3390/genes13101843_

Round 1
Reviewer 2 Report
The article is potentially interesting and informative. However, English grammar and style need extensive revision. Overall, mistakes in verb tenses, spelling, punctuation, and misuse of words make the manuscript difficult to read and not acceptable in the present form.
The expression analysis needs further elucidation. Particularly, stress treatments and sample collection should be described in detail.
In addition, the prediction of the interacting proteins would benefit from a more meaningful discussion, based on the possible functions of such proteins in plant biology rather than in human diseases.
